# STATE-SPACE MODEL FOR CAUSAL STRUCTURE LEARNING OF THE NONSTATIONARY VIDEO SEQUENCES

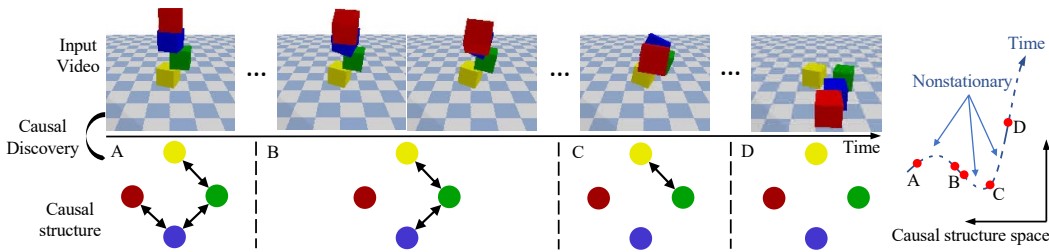

Figure 1: Illustration of nonstationary causal structures in physical systems. Causal structures can be represented as a graph, where edges indicate interaction between objects. Take the example of the supporting force in the block falling sequence, the graph changes over time, posing a challenge to video causal discovery methods.

## ABSTRACT

Nonstationary causal structures are prevalent in real-world physical systems. For example, the stacked blocks interact until they fall apart, while the billiard balls move independently until they collide. However, most video causal discovery methods can not discover such nonstationary causal structures due to the lack of modeling for the instantaneous change and the dynamics of the causal structure. In this work, we propose to capture the property of nonstationary for causal structure learning from video sequence. In particular, we leverage the state-space model (SSM) to formulate the change of the causal structure in a principled manner. Then, we use a recurrent model to sequentially predict the causal structure model based on previous observations to capture the nonstationary dynamic of the causal structure. We evaluate our method on two popular physical system simulation datasets with various types of multi-body interactions. Experiments show that the proposed achieves the state-of-the-art performance on both the counterfactual reasoning and future forecasting tasks.

## 1 INTRODUCTOIN

Causal reasoning from visual input is essential for intelligence systems in understanding the complex mechanisms in the physical world. For instance, autonomous vehicles need to infer the unseen causal structures on the road that drives the state evolution of other agents across time to anticipate future events better accordingly. One main obstacle in discovering such causal structures is the dynamic nature of events. In Figure 1, we illustrate the varying casual relationship in a simple multi-body system where the stacked blocks fall to the ground. In nonstationary video sequences, the causal structure can have abrupt changes and/or long-term dependencies, posing challenges for casual graphical models (CGM).

For the first challenge, most CGMs in video causal understanding can not handle abrupt causal relationship changes. For the second challenge, most CGMs in video causal understanding purely depend on the object state observations. That is the causal graph at time $t$ is conditionally independent from the causal graph at time $t-1$ given the object states' observations. Illustrated in Figure 1,

CGMs that can be represented as graphs can be modeled as a trajectory in the nonstationary video. In this work, we naturally extend the state-space model to the graph with time-leg edges in videos, *i.e.,* current objects' states are fully determined by previous states.

We summarize the contribution of this work as follows:

- We introduce the model to extend the state-space model framework to nonstationary video sequences.
- We achieve state-of-the-art performance on the task of counterfactual reasoning and the standard benchmark datasets CoPhy (Baradel et al., 2020).

## 2 METHODOLOGY

In this section, we present a state-space model based method for video causal discovery. We first give an overview of model architecture, as shown in Figure **??**, then dive into the components of our model.

### 2.1 PROBLEM FORMULATION

We factorize the joint probability of a temporal sequence into a sequential form:

$$p(\boldsymbol{x}^{1:T}; \theta) = p(\boldsymbol{x}^1; \theta) \prod_{t=2}^{T} p(\boldsymbol{x}^t | \boldsymbol{x}^{1:t-1}; \theta), \tag{1}$$

where $\theta$ is the model parameters to learn. This formulation makes it easy to do future forecasting by conditioning any unknown $\boldsymbol{x}^t$ on observed or previously predicted history $\boldsymbol{x}^{1:t-1}$. For simplicity, we decode multiple frames in an autoregressive way, i.e., at each timestep, we predict $\hat{\boldsymbol{x}}^t$ as the mode of $p(\boldsymbol{x}^t | x^{1:t-1}; \theta)$ and do further prediction conditioning on this prediction.

### 2.2 MODEL DESIGNS

Formally, given the observed $d$ agents in the scene from time 1 to $T$, a joint probability distribution $f(\boldsymbol{x})$ depict their state through time. In the context of Causal Graph Model (CGM) (Pearl et al., 2016), a directed acyclic graph (DAG) $\mathcal{G}$ with $dT$ nodes defines $f(\boldsymbol{x})$, where node $\boldsymbol{x}_j^t$ is associates with agent $j$ at time step $t$. Directed edges represents causal relationships. The distribution of agent states at time $t$ can be factorized as:

$$f(\boldsymbol{x}^t | \boldsymbol{x}^{1:t-1}; \theta) = \prod_{j=1}^{d} f(\boldsymbol{x}_j^t | Pa(\boldsymbol{x}_j^t); \theta), \tag{2}$$

where $Pa(\boldsymbol{x}_j^t)$ pertains to the set of parent nodes of $\boldsymbol{x}_j^t$ in $\mathcal{G}$. Eq. 2 implicitly hypothesizes the causal sufficiency (Peters et al., 2017), i.e., our work does not involve any hidden confounding elements. Also, we neither consider the instantaneous edges nor edges that go back in time in this work. Simply put, $Pa(\boldsymbol{x}_j^t) \subseteq \{\boldsymbol{x}_j^i\}_{i<t}$. This feature makes our causal graph fully identifiable in the context of video sequence as Li et al. (2020).

In order to properly capture the evolved connections between $Pa(\boldsymbol{x}_j^t)$ and $\boldsymbol{x}_j^t$, we introduce $\mathcal{M}^t$. Thus, the aforementioned likelihood function involving $\mathcal{M}^t$ arrives at:

$$f(\boldsymbol{x}^t | \mathcal{M}^t, \boldsymbol{x}^{1:t-1}) = \prod_{j} \tilde{f}\left(\boldsymbol{x}_j^t; \text{NN}(\mathcal{M}_j^t \odot \boldsymbol{x}; \phi_j^t)\right) \tag{3}$$

where NN and $\phi_j^t$ speak of the neural network and the associated parameters.

Learning $\mathcal{M}_j^t$ is tricky since this work does not assume any prior knowledge or access the groundtruth on it. To the end of tackling this problem, we invoke the state-space model (SSM) (**?**Yang et al., 2020; Huang et al., 2019; Hsieh et al., 2021). The SSM treats $\mathcal{M}_j^t$ as a latent variable

and learns it in a manner of variational inference. Specifically, we formulate the transition function of SSM by:

$$\begin{cases} h_t = f(h_{t-1}, \alpha_{t-1}, x_{t-1}) \\ \qquad\qquad \alpha_t \sim p(\alpha_t|h_t) \end{cases} \tag{4}$$

In this work, we consider a multivariate Gaussian to model $p(\alpha_t|h_t)$. The observation function can be written as following:

$$o_t \sim p(h_t, \mathcal{M}^t \sim \text{Bern}(\alpha^t), o_{t-1}) \tag{5}$$

## 2.3 LEARNING AND INFERENCE

Our training objective follows a manner of variational inference as follows:

$$\mathcal{L} = \sum_t \left( \underbrace{\mathbb{E}_{q(\alpha_t|x_t)} \log p(o_t|\alpha_t)}_{\text{Reconstruction}} - \underbrace{\text{KL}\big(q(\alpha_t|x_t)||p(\alpha_t|\alpha_{t-1})\big)}_{\text{KL Divergence}} \right) \tag{6}$$

$q(\alpha_t|x_t)$ refers to the posterior distribution of $\alpha_t$, which requires the observational sequence $x_{1:t}$ to calculate.

## 3 EXPERIMENT

### 3.1 EXPERIMENTAL SETUPS

**Downstream tasks and Datasets.** We conduct experiments to understand the efficacy of our proposed method in terms of discovering the causal structure to estiamte the object dynamics across time. More specifically, the counterfactual reasoning of a video sequence is selected to demonstrate this point.

Counterfactual Reasoning of a video sequence is formalized as follows (Baradel et al., 2020): During training, we first infer the causal structure upon a set of visual observations. The objective is to reason the counterfactual outcome given the modified initial object state. The Counterfactual Physics benchmark (CoPhy) (Baradel et al., 2020) dataset contains two types of sequences, observational and counterfactual. The latter sequence is built upon changing the initial object state from the observations with other factors ((such as inertia, gravity or friction)) untouched. CoPhy comprises three physical scenarios in total: BlockTowerCF, BallsCF and CollisionCF. Each scenario provides the 3D positions of all objects in the scene. BlockTowerCF also includes a binary label for stability. On each dataset, we directly use the extracted visual features from video frames in the previous state-of-the-art methods. Below are the details.

*Visual Features.* For observation $x^t$, we use the extracted visual features from input videos to improve the model performance. For a fair comparison on CoPhy, we adopt the identical experimental protocols in (Baradel et al., 2020) to examine the generalizability of . We train and test with 4 objects on BlockTowerCF and BallsCF. The experiments on CollisionCF utilize all types of objects (spheres and cylinders) for both training and test.

*Model Architectures.* We append on a two-layer GRU to instantiate the transition model, and a three-layer MLP to parameterize the observation model. At the time instance $\tau$, $\alpha_\tau$ is then reshaped to a set of $d \times d$ matrices for $\mathcal{M}^t$. Notably, we zero-padded these matrices to ensure there exists $t - 1$ individual matrix in total per time instance for backpropagation. For the faster learning convergence, we place an instance normalization layer before each ReLU activation in the MLP model and use the sigmoid activation for the final output to make it a probability value.

*Learning and Inference.* In our experiments, RMSProp optimizer (Goodfellow et al., 2016) are employed with the learning rate initialized at $8 \times 10^{-5}$ . Our implementation uses PyTorch. The experiments are executed on four Nvidia GeForce TITAN XPs, with 48 GB of memory in total.

**Evaluation Metrics.** Since none of the aforementioned datasets provide annotations for the causal graphical model, we gauge model performance by the observed object dynamics which is generated from the unobserved causal structure. Thus the ideal metrics should rely on object states, i.e., coordinates and stability. In particular, we aim to understand how close the outcomes can approximate the

Table 1: Quantitative results on three physical scenarios of the CoPhy dataset and Fabric dataset. We compare with state-of-the-art methods: CoPhyNet (Baradel et al., 2020), and V-CDN (Li et al., 2020). Non-existent experiments are marked by hyphens.

| | CoPhy Dataset (Baradel et al., 2020) | | | | |
| | BlocktowerCF | | | CollisionCF | |
| | MSE ↓ | NLL ↓ | Acc. ↑ | MSE ↓ | NLL ↓ |
|---|---|---|---|---|---|
| CoPhyNet | 0.49 | 8.52 | 73.8 | 0.22 | 6.97 |
| (Ours) | **0.45** | **7.97** | **75.1** | **0.20** | **6.70** |

ground truth. To this end, we calculate the mean square error (MSE) and the negative log-likelihood (NLL) on coordinates of objects between ground-truth and prediction. NLL is the average negative log-likelihood between a ground truth trajectory distribution determined by a kernel density estimate and the predicted trajectory. In addition, the stability classification accuracy are used for our experiment on BlockTowerCF. Lower NLL and MSE and higher accuracy are preferred.

## 3.2 DISCUSSIONS

As per comparing methods, we are primarily interested in assessing our versus the leading studies on estimating agent states in a video sequence in the context of learning CGM. More specifically, CoPhyNet (Baradel et al., 2020), which achieves cutting-edge results on the CoPhy benchmark is selected. CoPhyNet summarizes the problem with a given causal structure to handle the object dynamics over time and approache object interactions with fully-connected graph convolution (**??**). To the best of our knowledge, this method is the most relevant ones to ours.

It can be seen in Table 1 that out model consistently beat CoPhyNet. It demonstrates the necessity of capturing nonstationary causal structures for counterfactual reasoning of the video sequences.

## 4 RELATED WORK

The relevant literature in the computer vision community has accumulated several efforts to tackle down the challenges of video modeling and prediction. Nevertheless, one topic that had enjoyed recent success is reasoning objective dynamics in a video sequence. A line of research attempts to solve this task by modeling the correlations in a spatio-temporal context, such as (Yi* et al., 2020; Chen et al., 2021; Bakhtin et al., 2019; Qi et al., 2021; Zhang et al., 2021). However, focusing on modeling the dependencies substantially might not suffice to offer clear interpretations of object dynamics as we humans do. Addressing this issue, the authors of (Baradel et al., 2020) and (Li et al., 2020) try to make efforts to introduce causal knowledge (Schölkopf et al., 2021; Bengio et al., 2020; Runge et al., 2019) to this task. A few works adapt various topics into such a context. Whereas neither of them is able to fully uncover the causal structure underlying the video sequences: CoPhyNet (Baradel et al., 2020) derives an alternative output based on a known causal graph; VCDN (Li et al., 2020) focus on recovering the stationary causal structures from the video. Instead, our proposed method apply the state-space model to capture nonstationary causal structures.

## 5 CONCLUSION

In this paper, we propose an state-space model based framework for learning the causal structure of the video sequences. Our method differs from works the literature in that it introduces the SSM discover the nonstationarity causal structure for understanding the object dynamics in video sequences. Experiment results justify that our delivers better performance in the counterfactual reasoning with respect to prior works. One direction is to loose the sufficiency assumption and involve the confounding elements to our framework to enable discovering the causal relationships in real-world applications.

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
