# OpenReview forum: "Intervention-based Recurrent causal Model for Nonstationary Video Causal Discovery"
_ICLR.cc/2022/Workshop/OSC — Submitted to ICLR2022 OSC _

### Official Review · Reviewer_VWDn · 2022-03-06

**Rating:** 2
**Confidence:** 2

**Review:**

This paper extends existing casual learning methods to the nonstationary video cases, where the casual structures are changing throughout time. The proposed model leverages a recurrent model GRU to extract temporal information, which is intuitively correct because this will capture the changes of casual graphs over time. Since there is no GT casual graph as direct supervision, the authors employ state-space model and perform variational inference on it. The proposed model achieves superior performance compared to one previous work on one dataset

Detailed comments:
- I completely agree that the evolution of casual structure in nonstationary videos is indeed an important and new problem. The usage of GRU to tackle this problem seems reasonable to me
- However, since there is no GT casual graph in the dataset, the author doesn't visualize the casual graph learned in any case. It's unclear to me whether SSM and eq3 can really help the model learn the correct underlying casual graph
- There are many typos/writing errors in the paper. At the beginning of Section 2, the referred model architecture figure is missed. In Table 1, the baseline V-CDN is mentioned but missed in the table

Nonetheless, I still believe the discovery of casual structure in nonstationary videos is an important paper. So I recommend acceptance of this paper to the workshop.

---

### Official Review · Reviewer_B4rL · 2022-03-16
**Simple idea to learn causal graph structure for video causal models, but significant discrepancies in experimental reporting.**

**Rating:** 1
**Confidence:** 3

**Review:**

The paper investigates the problem of causal discovery, more specifically the challenge in video causal discovery systems to model instantaneous and dynamical change in the underlying causal graph in the intuitive physics domain (bouncing balls or tower of stacked blocks).  The main idea is to use a state-space model wherein the adjacency matrix of the causal graph is modeled as a latent variable and learned using the standard ELBO objective.

Pros:
Model has been evaluated on the benchmark CoPhy dataset and against a strong baseline method designed for the same benchmark.
The research question under consideration is highly relevant to the workshop theme.

However, there are significant issues with regards to the experiments and results reported as described below.

Cons:

The experimental setup and results section is poorly recorded with many discrepancies between what is claimed versus what is reported. In “Downstream tasks and Datasets” the authors mention that they use all 3 datasets (i.e. BlockTowerCF, BallsCF and  CollisionCF) from the CoPhy benchmark. Further, the caption of Table 1 reads “Quantitative results on three physical scenarios of the CoPhy dataset and Fabric dataset. We compare with state-of-the-art methods: CoPhyNet (Baradel et. al 2020) and V-CDN (Li et. al 2020). Non-existent experiments are marked by hyphens”. However, the Table 1 only compares the proposed model against CoPhyNet and only on 2 datasets — BlockTowerCF and CollisionCF. No results have been reported on BallsCF and the Fabric dataset. Results for the V-CDN baseline are also missing. It’s unclear what the last sentence of the caption means since there are no hyphenated results shown in Table 1.

Writing: The writing and presentation could be significantly improved to enhance the clarity of exposition and readability. Below are some typos/errors I found:

- “property of nonstationary” -> “property of non-stationarity”

- “… show that the proposed achieves” -> “… show that the proposed model/method achieves”

- “… is essential for intelligence systems in understanding the complex mechanisms in the physical world.“ missing citation.

- “… as shown in Figure ??” — missing reference (latex error)

- “… predict $\hat{x}^t$ is the mode of $p(x^t | x^{1:t-1} ; \theta)$ and do further prediction conditioning on this prediction” - sentence construction could be improved.

- “… given the observed $d$ agents ” — what is an agent in this context?

- “… node $x_j^t$ is associates” -> “… node $x_j^t$ is associated”

- “… we invoke the state-space model (SSM) (?, Yang et. al …)” — missing reference/latex error

- In eq. 4 $\alpha_t \sim p(\alpha_t | h_t)$ -> unclear notation. Further, the time variable ’t’ is alternating used as both a superscript and
subscript in different situations which is difficult to follow for the reader.

- “… causal structure to estimate” -> “… causal structure to estimate”

- “… experimental protocols in (Baradel et. al 2020) to examine the generalizability of” — sentence incomplete.

- “3.2 DISCUSSIONS” -> “3.2 DISCUSSION”

- “… interested in assessing our versus” -> “… interested in assessing our model/method versus”

- “… dynamics over time and approache object interactions with fully-connected graph convolution (??)” -> “… dynamics over time and approaches object” + missing citation

- “… seen in Table 1 that out model” -> “… seen in Table 1 that our model”

- “… propose an state-space model” -> “… propose a state-space model”

- “… justify that our delivers” -> “justify that our model/method delivers”

---

### Decision · Program_Chairs · 2022-03-19

**Decision:**

Reject

**Comment:**

The reviewers agree that this paper addresses an important problem with high relevance to the workshop, but several concerns were raised about the quality of the presentation/writing and discrepancies in the experimental evaluation. These concerns need to be addressed before this paper is ready for sharing with a wider audience and we recommend submission to a later conference or workshop (as the timeline for preparing camera-ready revisions for workshop papers is quite tight).